# Boosting Expression of a Specifically Targeted Antimicrobial Peptide K in *Pichia pastoris* by Employing a 2A Self-Cleaving Peptide-Based Expression System

**DOI:** 10.3390/antibiotics13100986

**Published:** 2024-10-18

**Authors:** Yunhui Zhu, Yuwen Li, Yuxin Fang, Mingyang Hu, Lu Zhao, Mingrui Sui, Na Dong

**Affiliations:** Laboratory of Molecular Nutrition and Immunity, College of Animal Science and Technology, Northeast Agricultural University, Harbin 150038, China; b210501012@neau.edu.cn (Y.Z.); lyw7992021@163.com (Y.L.); yuxinfang168@163.com (Y.F.); humingyang0505@163.com (M.H.); zl1654501466@163.com (L.Z.); smr180928neau@163.com (M.S.)

**Keywords:** selective targeted antimicrobial peptide, *Pichia pastoris*, 2A self-cleaving peptide, fermentation condition optimization

## Abstract

**Background/Objectives:** The current epidemic of drug-resistance bacterial strains is one of the most urgent threats to human health. Antimicrobial peptides (AMPs) are known for their good activity against multidrug resistance bacteria. Specifically targeted AMPs (STAMPs) are a fraction of AMPs that target specific bacteria and maintain the balance of the healthy microbiota of a host. We reported a STAMP Peptide K (former name: peptide 13) for E. coli. The aim of this study was to effectively produce peptide K using methylotrophic yeast *Pichia pastoris*. **Methods:** Three inserts (sequence of peptide K (K), two copies of peptide K fused with 2A sequence (KTK), and two copies of peptide K fused with 2A and an extra α mating factor (KTAK)) were designed to investigate the effect of the number of repeats and the trafficking of peptide on the yield. **Results:** The yield from KTK was the highest—more than two-fold higher compared with K—implying the role of the 2A sequence in heterologous peptide expression apart from the co-translation. Then, the fermentation condition for KTK was optimized. The optimized yield of KTK was 6.67 mg/mL, suggesting the efficiency of the expression system. Selectivity, antibacterial activity, biocompatibility, and the stability of the fermentation product were equivalent to the chemically synthesized peptide. The actional mechanism of the fermentation product included membrane permeabilization and ROS induction. **Conclusions:** Together, our work provided a new perspective to augment the yield of the antimicrobial peptide in the microbial system, building a technological foundation for their large-scale production and expanding the market application of AMPs.

## 1. Introduction

The current epidemic of drug-resistance bacterial strains owning to the abuse and excessive use of antibiotics is one of the most urgent threats to human health [1]. Antimicrobial peptides (AMPs) are a group of small peptides known for their good activity against multidrug resistance bacteria [2]. Conventional antibiotics mainly act on targets, i.e., bacterial ribosomes, transpeptidases, and topoisomerases [3]. These molecules are easily mutated and therefore escape from the action of antibiotics, whereas AMPs act via direct killing by rupturing the bacterial membrane and/or suppressing the activities of multiple molecular targets [4]. The action of AMPs is swifter and less likely to induce bacterial resistance. Hence, AMPs have been extensively investigated by researchers as drug candidates in an attempt to limit the spread of antibiotic resistance. Specifically targeted AMPs (STAMPs) are a fraction of AMPs that target specific bacteria without harming the benign ones and upsetting the balance of a healthy microbiota and the health of a host [5]. We reported a STAMP K (former name: peptide 13) for *E. coli* [6]. Peptide K was designed by conjugating a phage-displayed peptide against *E. coli* with a rationally designed broad-spectrum AMP. K exhibited potent antimicrobial activity and selectivity against *E. coli*. The actional mechanism included the rupture of the cytoplasmic membrane and further cytosol leakage. In addition, K presented low cytotoxicity and desirable salt, serum, acid, and heat stability. Generally, K is a peptide that presents great potential to replace antibiotics.

One factor impeding the widespread application of AMPs is the high cost of their production [7]. Rationally designed peptides are normally obtained via two ways: direct chemical synthesis and biosynthesis [8]. Chemical synthesis is expensive and the synthetic difficulties soar as the scale of production increases. The biological method is more suitable for peptide production in large scale, but the activity and stability of secreted AMPs varies [7]. Recombinant production is a relatively simple and economic biological method to produce peptides by incorporating peptide genes in a specific expression system. Among numerous expression systems for expressing foreign peptides, methylotrophic yeast *Pichia pastoris* (*P. pastoris*) is widely used [9,10]. The advantages of *P. pastoris* include (1) a well-characterized genetic background; (2) the genetic tractability and good genetic stability of recombinant elements even in large scale production; (3) a short growth cycle and high peptide expression with a low nutrient requirement; and (4) simple downstream processing, i.e., the separation and purification of the exogenous peptide in a later stage. These features allow the achievement of large amounts of peptides with relative technical facility at a lower cost and high speed. Expressing tandem repeats has been a strategy to enhance the expression level of recombinant products [11]. Conventionally, the simultaneous expression of a multi-copy gene cassette in *P. pastoris* requires separate promoters and terminators because of the absence of a polycistronic gene expression system. But, the overall expression efficiency may be affected due to the competition between transcription factors, influence from upstream genes, and the overall size of the transcript [12]. The 2A self-processing peptide offered an alternative to boost the production of the exogenous peptide. The 2A self-processing peptide can function as a single transcriptional unit to directly interfere with the translational process and spontaneously initiate the synthesis of a new protein [13]. The success of the 2A self-processing peptide in boosting protein expression has been reported in numerous studies [13,14,15]; however, the success of the 2A peptide in peptide expression is relatively lower, especially AMP expression in *P. pastoris*.

This work aims to develop a strategy to cost-effectively express antimicrobial peptide K. Recombinant *P. pastoris* harboring a two-copy cassette of K with or without an α mating factor was constructed via 2A peptide assembly. The alpha factor is a signal peptide, sorting protein secretion from cytosol to the extracellular matrix and impacting the secretion efficiency and yields of the targeted product [16]. The induction was optimized with fermentation duration, pH, and methanol concentration. The yield and antimicrobial activity and stability of the fermentation product were assessed. The actional mechanism of the product was elucidated.

## 2. Results

### 2.1. Construction of P. pastoris Expressing Recombinant Antimicrobial Peptide K

The amino acid sequence of antimicrobial K, a 2A peptide, and an α mating factor was listed in Table 1. The nucleic acid sequences of K alone—two-fold K linked by 2A assembly (KTK)—and two-fold tandem plus α mating factor linked by 2A assembly (KTAK) were optimized according to the bias of *P. pastoris*. Optimized nucleic acid sequences for K, KTK, and KTAK were listed in Appendix A. An EcoR I recognition sequence (GAATTC) was inserted at the N-terminus. A termination codon (TAA) and a Not I recognition sequence (GCGGCCGC) were inserted at the C-terminus of target genes (Figure 1a).

The construction of recombinant pPIC9K-K, pPIC9K-KTK, and pPIC9K-KTAK was assessed with agar gel electrophoresis (Figure 1b). Constructed pPIC9K plasmids were digested with a restriction enzyme (EcoR I and Not I). As expected, 91 bp, 235 bp, and 502 bp fragments were observed from recombinant pPIC9K-K, pPIC9K-KTK, and pPIC9K-KTAK, indicating the success of construction.

Plasmid pPIC9K was transformed into GS115 by electroporation and positive GS115 transformants were screened with MD and MM plates (Figure 1c and Appendix A). Monoclonal recombinant GS115 was picked and assessed by PCR using a universal primer for the Aldehyde Oxidase I (*AOX1*) promoter [17], a promoter for methanol inducible protein expression in *P. pastoris*. PCR agar gel electrophoresis suggested longer products in recombinant GS115 compared to the GS115 containing empty plasmid (Figure 1d). Gene sequencing confirmed the success of transformation (Appendix A). Together, pPIC9K-K, pPIC9K-KTK, and pPIC9K-KTAK were successfully constructed and transformed into GS115 cells.

### 2.2. Expression of Peptide K from Recombinant P. pastoris

A monoclonal colony of GS115 containing empty pPIC9K was seeded in BMGY, and the growth performance of GS115 was firstly monitored. The log phase of GS115 containing pPIC9K plasmid was around 4–6 h of inoculation (Figure 2a). The process of fermentation was firstly assessed by the wet weight of GS115 (Figure 2b). The result revealed that the wet weights reached their peaks at 72 h. The fermentation products in the culture supernatant were analyzed with Tricine-SDS-PAGE, with the potential target band framed with dash line (Figure 2c). The size of the products from K, KTK, and KTAK should be 3.2 to 5.1 KDa. We failed to observe the bands around 3.2–5.1 KDa, but the supernatant from GS115 containing pPIC9K-K, pPIC9K-KTK, and pPIC9K-KTAK presented more bands compared to the supernatant from GS115 containing empty pPIC9K. This suggested that the supernatant from pPIC9K-K, pPIC9K-KTK, and pPIC9K-KTAK contained more products compared to the supernatant from empty pPIC9K, and that peptide K should be successfully expressed. The reason why the bands were not at the supposed position may be the interference of the peptide’s charge and/or the existence of Pro residue on the migration rate [18,19,20]. To test our inference, we assessed the antimicrobial activity of fermentation supernatants (Figure 2d). The supernatant effectively reduced the viability of *E. coli*. We then assessed the amount of total proteins and targeted peptides in the supernatant (Table 2). As we expected, the total and targeted product was highest in the supernatant from GS115 containing pPIC9K-KTK. Interestingly, the total product from pPIC9K-KTAK was higher than that from pPIC9K-K, while the target product from pPIC9K-KTAK was lower than that from pPIC9K-K. The pPIC9K-KTK expression system was selected for the further optimization of the fermentation process.
$$C_{\mathrm{Target}}=\frac{C_{Elution}\;\times \;V_{Elution}}{V_{supernatant}}$$

### 2.3. Optimizing the Fermentation Conditions of Recombinant P. pastoris GS115

The process of fermentation was assessed with the wet weight of recombinant GS115 and the production of proteins. We firstly investigated the effect of the induction duration on the production (Figure 3a). The potential target band should be near 6.5 KDa, deducing from the WB with the His tag antibody for the supernatant (Appendix A) and the Tricine-SDS-PAGE for the purified product (Figure 4a). The results suggested that both the wet weight of GS115 and the protein production peaked at 72 h. Then, the optimal concentration of methanol was investigated (Figure 3b). The wet weight and protein production peaked with 1.0% methanol. Last, the optimal pH value was investigated (Figure 3c). The optimal pH value is 6.0. Together, the optimal fermentation parameters for recombinant GS115 were 72 h, 1.0% methanol, and pH 6.0. The concentration of the purified peptide K was 6.67 mg/mL, determined with 1 mL of obtained solution.

### 2.4. The Bioactivity and Actional Mechanism of Recombinant Peptide K

The bioactivity and actional mechanism of recombinant peptide K were assessed with a purified product (Figure 4a). The antibacterial activity of the purified recombinant peptide K against different bacteria was firstly examined (Table 3). Recombinant peptide K exhibited good selectivity against *E. coli* strains, with an average MIC of 14.6 μg/mL. Recombinant peptide K showed no hemolytic activity at a concentration of 200 μg/mL (Figure 4b). Melittin has also been used as positive control by virtue of its representative bioactivity and actional mechanism. We then evaluated the antibacterial activity of recombinant peptide K in the presence of salts, pH values, serums, and high temperature (Table 4). Recombinant peptide K exhibited good stability in the presence of K^+^, NH_4_^+^, Ca_2_^+^, Zn_2_^+^, and Fe_3_^+^. The antibacterial activity of recombinant peptide K did not vary much under different pH values, and the antibacterial activity did not alter in the presence of high temperature. The presence of serum increased the minimum inhibitory concentration of recombinant peptide K. Last, we examined whether the actional mechanism of recombinant peptide K remained the same. As expected, the results suggested that recombinant peptide K killed bacteria using membrane permeabilization and ROS induction (Figure 4c,d).

## 3. Discussion

The spread of antibiotic resistance among pathogens poses a threat to public health [3]. The antimicrobial peptide has been deemed as one of the most attractive alternatives to conventional antibiotics because of their potent antimicrobial activities, low resistance, and low residue [2,4,21]. Specifically targeted antimicrobial peptides are capable of specifically killing specific pathogens without disturbing the normal flora, and gaining research interests in regard to the importance of the ecological balance of normal microbial communities [5]. We have reported a specifically targeted antimicrobial peptide K that can effectively kill *E. coli*, but spare probiotics, presenting potential in substituting antibiotics [6]. To date, AMPs are mainly obtained with the chemical synthesis method. The imbalance between the cost and the large-scale production of antimicrobial peptides limits their widespread application. AMPs with natural amino acids can be genetically encoded. Therefore, the microbial recombinant expression system offers an opportunity to produce AMPs in large quantities with a relatively low cost and technical facility. *P. pastoris*, methylotrophic yeast, is a preferred host for recombinant expression, because (1) *P. pastoris* can rapidly increase their biomass without the requirement of expensive media and equipment; (2) enable the target genes to integrate into the yeast genome directly, which allows the stable expression of target genes; and (3) produce heterologous products into the culture medium directly, making it easy for separation and purification to occur in the later stage [9]. Our work focused on optimizing the production of peptide K in the cells of methylotrophic yeast *P. pastoris*. A common strategy to enhance the yield includes increasing the copies of target genes in the fusion construction, screening for efficient integrant, and optimizing fermentation conditions [22]. The 2A peptide sequence can lead to a ribosomal skip. When a ribosome encounters 2A within an open reading frame, the nascent protein is released from the ribosome, and the ribosome immediately resumes the translation of the downstream sequences. Due to this property, the 2A sequence has been widely used to co-express multiple different proteins derived from a fusion precursor in a single cell [23]. Yan et al. [24] reported that the product upstream of 2A was targeted at the endoplasmic reticulum while the downstream ones were targeted at the cytosol. In the present study, sequences of peptide K, two copies of peptide K fused with 2A sequence, and two copies of peptides K fused with 2A sequence plus an extra α mating factor (the most widely used secretion signal sequence sorting recombinant protein secretion from cytosol to extracellular matrix) were designed, optimized, and transformed to *P. pastoris* GS115. The yield efficiency of these inserts was assessed, and the fermentation conditions of recombinant GS115 producing the highest yield was optimized. We hope our work can help to build a technological foundation for the large-scale industrial production of antimicrobial peptides and expand their market.

The wet weight of GS115 containing pPIC9K-K, pPIC9K-KTK, and pPIC9K-KTAK peaked at 72 h after the induction of fermentation, then reduced at 96 h after induction. One reason for this may be that, as the fermentation time extended, the density of the cell increased, nutrients in medium were deprived, cells entered the aging state, and the biomass of *P. pastoris* reduced. Another reason may be that *P. pastoris* are able to secrete enzymes that can degrade heterologous proteins. At 96 h after induction, the degradation overwhelmed the secretion, resulting in a reduction in the wet weight and yield. Among all the inserts, the total yield from GS115 containing pPIC9K-K was the lowest, and that from GS115 containing pPIC9K-KTK was the highest. The specific yield was assessed by determining the specific bands in pPIC9K-K, pPIC9K-KTK, and pPIC9K-KTAK. The yield from pPIC9K-KTK was still the highest, while the yield from pPIC9K-KTAK was the lowest. This confirmed that tandemly arraying expression cassettes with the 2A sequence augmented the yield of peptide K. The α mating factor is responsible for sorting the posttranslational translocation of heterologous products across the endoplasmic reticulum membrane, and is the most commonly used secretion signal for *P. pastoris* [25]. Yet, the lack of success of protein secretion in the presence of the α mating factor signal has been reported [26]. The secretion of heterologous proteins can be influenced by the stress response in the endoplasmic reticulum membrane, protein trafficking, and sorting from the endoplasmic reticulum to the Golgi apparatus and then to the extracellular matrix. One reason why the inclusion of an extra α mating factor reduced the yield might be that the increased translocation resulted in a stress response in endoplasmic reticulum and reduced the yield. In addition, Idiris et al. [27] reported a hold of rapidly folding proteins in cytosol in the presence of α mating factor. Peptide K is short in length and may undergo a rapid fold in the cytosol; therefore, the increase in the α mating factor secretion signal resulted in a poor yield.

The fermentation condition of pPIC9K-KTK was then optimized. Still, the wet weight and the amount of total protein peaked at 72 h after induction. 72 h after induction was selected as the optimal induction time point for the fermentation of pPIC9K-KTK. *P. pastoris* utilized normal carbon sources (i.e., glycerol) to increase cell density [28]. High cell density is a prerequisite for the high expression of heterologous proteins. Methanol is required for the expression of heterologous proteins [17]. Therefore, both carbon sources need to be present. The optimal methanol concentration for KTK is 1%. The yield was lower in the presence of 1.5% and 2.0% methanol. This may be because the toxicity of the high concentration of methanol, due to the accumulation of formaldehyde and hydrogen peroxide. The pH of the medium affected the growth state of *P. pastoris* and the expression of the heterologous product. The yield was highest with pH = 6.0. Together, the fermentation efficiency for pPIC9K-KTK was highest at 72 h after induction with 1% methanol and pH 6.0. The follow-up assessment of the bioactivity, biocompatibility, stability, and actional mechanism of the fermentation product was equivalent to that of the chemically synthesized peptide K, suggesting the efficiency of the expression system.

## 4. Materials and Methods

### 4.1. Bacterial Strains, Plasmids, and Reagents

The strains for the specific targeted antimicrobial activity included *E. coli* ATCC 078, *E. coli* ATCC 25922, *E. coli* UB 1005, *E. coli* K99, *E. coli* K88, *E. coli* 987p, *S. aureus* ATCC 29213, *S. aureus* ATCC 25923, *S. epidermidis* ATCC 12228, *P. aeruginosa* ATCC 27853, *S. typhimurium* ATCC 7731, *L. plantarum* 8014, and *L. rhamnosus* 7469. The above strains were preserved by the laboratory. The *P. pastoris* strain GS115, pPIC9K plasmid, Geneticin^®^, and His Tag antibody were bought from Thermo Fisher Co., Ltd. (Waltham, MA, USA). The restriction endonucleases EcoR I, Not I, and Sac I were bought from Beyotime Biotech (Shanghai, China). The plasmid extract kit and the Gel extraction kit were from Vazyme Biotech (Nanjing, China). The Tricine-SDS-PAGE kit and the fast silver stain kit were from Solarbio Co., Ltd. (Beijing, China). The purity of all the other chemical reagents used was of analytic grade.

### 4.2. Plasmid Construction

Full-length sequences were synthesized by Sangon Biotech Co., Ltd. (Shanghai, China). The synthesized sequences were inserted into the pPIC9K between the EcoR I and Not I downstream of the α-factor signal peptide. The construction of plasmid pPIC9K containing target sequences was entrusted to Sangon Biotech Co., Ltd.

### 4.3. Transformation of P. pastoris GS115

The linearization of plasmid was performed with *Sac I* restriction endonuclease. Then, the linear plasmids were electroporated into GS115 growing in the log phase. The possible inserts were screened with the agar plate containing Minimal Dextrose (MD)/Minimal Methanol (MM) and Geneticin^®^ (G418; 100 μg/mL). The genome of *P. pastoris* was extracted with a DNA extraction kit. The insertion was verified by colony PCR. The primers used were 5′-GACTGGTTCCAATTGACAAGC-3′ and 3′-GGCAAATGGCATTCTGACAT-5′. The amplified products were then subjected to agarose gel electrophoresis. The separated DNA was retrieved with a Gel extraction kit. The sequencing of retrieved DNA was entrusted to Comate Bioscience Co., Ltd. (Jilin, China). The empty vector was used as the negative control.

### 4.4. Induced Expression of Recombinant Antimicrobial Peptide K and the Optimization of Fermentation Conditions

The general induction procedures were as follows. The growth of *P. pastoris* in buffered glycerol complex medium (BMGY) was determined every hour by measuring the optical density at 600 nm. For induction, positive transformants were cultivated in the BMGY to the log phase and then induced in buffered methanol-complex medium (BMMY). Methanol was added to a certain final concentration every 24 h. At the end of the induction, the fermentation supernatant was collected, centrifuged (10,000 rpm, 4 °C, 10 min), and analyzed. The presence of the recombinant peptide in the supernatant was assessed with WB using the His Tag antibody. The trend of protein expression was determined with Tricine-SDS-PAGE. Bands were visualized with a fast silver stain kit. The concentration of the protein in the fermentation supernatant was determined with a Bradford assay.

For the optimization for the length of the fermentation duration, transformants were induced in buffered methanol-complex medium (pH = 6.0) with 1% (*v*/*v*) methanol for 24, 48, 72, and 96 h.

For the optimization for the concentration of methanol, transformants were induced in buffered methanol-complex medium (pH = 6.0) with 0.5%, 1%, 1.5%, and 2.0% (*v*/*v*) methanol for 72 h. 100% methanol was added to obtain a desired final concentration every 24 h.

For the optimization for the medium pH, transformants were induced in buffered methanol-complex medium (pH = 5.5, 6.0, 6.5, and 7.0) with 1% (*v*/*v*) methanol for 72 h. The initial pH of the culture medium was adjusted using H_3_PO_4_ and KOH.

### 4.5. Purification of Recombinant Antimicrobial Peptide K

Purification of K was performed with a nickel column. The total protein in the fermentation supernatant was extracted with 75% ammonium sulfate overnight. Extracted protein was obtained by centrifugation (12,000 rpm, 10 min, 4 °C), redissolved in binding buffer (20 mM Imidazole, 20 mM Tris-HCl pH = 8.0, 500 mM NaCl), and dialyzed with an ultra centrifugal filter. The column was balanced with binding buffer, and redissolved protein was added to the column containing Ni NTA filler, and combined with filler overnight. Elution buffer (500 mM Imidazole, 20 mM Tris-HCl pH = 8.0, 500 mM NaCl) was added to the column. The eluent was dialyzed with 1000 Da MWCO, and analyzed with Tricine-SDS-PAGE (Beijing, China).

### 4.6. Antibacterial Activity of Recombinant Antimicrobial Peptide K

The below protocols were applied to all the test bacteria used for the antimicrobial assessment for the determination of the antibacterial activity of the fermentation supernatant. The supernatant was collected and mixed with the same volume of PBS. The diluted supernatant was then mixed with the same volume of bacteria and incubated at 37 °C. After 2 h incubation, the 10 μL culture was aspirated and spread on an MHA plate. The antibacterial activity of the supernatant was determined with a colony forming analysis.

For the determination of the minimum inhibitory of purified products, serially diluted products were mixed with different bacteria (10^5^ CFU/mL). Then, the mixtures were incubated at 37 °C for 16–18 h. The minimum inhibitory concentration was recorded as the lowest concentration producing no visual turbidity. Bacteria incubated with PBS alone was selected as the positive control.

### 4.7. Hemolytic Activity of Recombinant Antimicrobial Peptide K

The hemolytic activity of the recombinant peptide K was performed with human red blood cells. All the procedures were approved by the Ethics Committee of Northeast Agricultural University Hospital, with written informed consent obtained from all the volunteers. Collected hRBCs were incubated with serially diluted fermentation supernatant and incubated at 37 °C for 1 h. The absorbance was measured at 570 nm. The cells treated with PBS or 0.1% Triton X-100 were employed as negative and positive controls. The hemolysis rate was calculated with the following formula:$$Hemolysis\;rate\left(\%\right)=\frac{{OD}_{570}\left(sample\right)-{OD}_{570}(negative)}{{OD}_{570}\left(positive\right)-{OD}_{570}(negative)}$$

### 4.8. Salt, Serum, Temperature, and pH Tolerance of Recombinant Antimicrobial Peptide K

The tolerance of peptide K was assessed by measuring the MICs of the peptide against *E. coli* K88 under different salt, serum, temperature, and pH conditions. In the present study, the final salt concentrations were as follows: 150 mM NaCl, 4.5 mM KCl, 6 μM NH_4_Cl, 8 μM ZnCl_2_, 1 mM MgCl_2_, 2 mM CaCl_2_, and 4 μM FeCl_3_. The final serum concentrations were 12.5%, 25%, and 50%. The pH of the MIC system was adjusted with 1 mM NaOH and 1 mM HCl.

### 4.9. Outer Membrane Permeability

The outer membrane permeability was evaluated by the uptake of N-phenyl-1-naphthylamine. Bacteria in the logarithmic growth phase was diluted to 107 CFU/mL and mixed with N-phenyl-1-naphthylamine (final concentration: 10 μM) in 4-(2-hydroxyeerhyl) piperazine-1-erhanesulfonic acid buffer (5 mM, pH 7.2, containing 5 mM glucose). The peptide was added after 30 min incubation at room temperature. The fluorescence intensity was measured with an excitation wavelength of 350 nm and emission of 420 nm.

### 4.10. Reactive Oxygen Species Generation

The production of ROS was assessed using 2′,7′-dichlorodihydrofluorescein diacetate. Bacteria were diluted to 4 × 108 CFU/mL with PBS and mixed with a serially diluted peptide. ROS production was assessed with an excitation wavelength of 488 nm and an emission of 525 nm.

### 4.11. Statistics

Data were analyzed with one-way analysis of variance and the Duncan test, using GraphPad Prism 9 software. Data were expressed as mean ± standard error. *p* < 0.05 suggests a difference.

## 5. Conclusions

The aim of this study was to effectively produce a specifically targeted antimicrobial peptide K by methylotrophic yeast *P. pastoris*. Three inserts (sequence of peptide K (K), two copies of peptide K fused with 2A sequence (KTK), and two copies of peptide K fused with 2A and an extra α mating factor (KTAK)) were designed to investigate the effect of the number of repeats and the trafficking of peptide on the yield. The yield from KTK was the highest at more than two-fold higher compared with K, implying the role of the 2A sequence in the heterologous peptide yield apart from the co-translation. The yield from KTAK was lower compared to K, suggesting the negative role of the extra α mating factor. Then, the fermentation condition for KTK was optimized. The optimized yield of KTK was 6.67 mg/mL, suggesting the efficiency of the expression system. The selectivity, antibacterial activity, biocompatibility, and stability of the fermentation product were equivalent to the chemically synthesized peptide K. The actional mechanism of the fermentation product included membrane permeabilization and ROS induction. Together, our work provided a new perspective to augment the yield of the antimicrobial peptide in the microbial system. We hope our work can offer a viable means/theoretical guidance for the scale production of AMPs and expand the market of AMP as an alternative for conventual antibiotics.

## Figures and Tables

**Figure 1 antibiotics-13-00986-f001:**
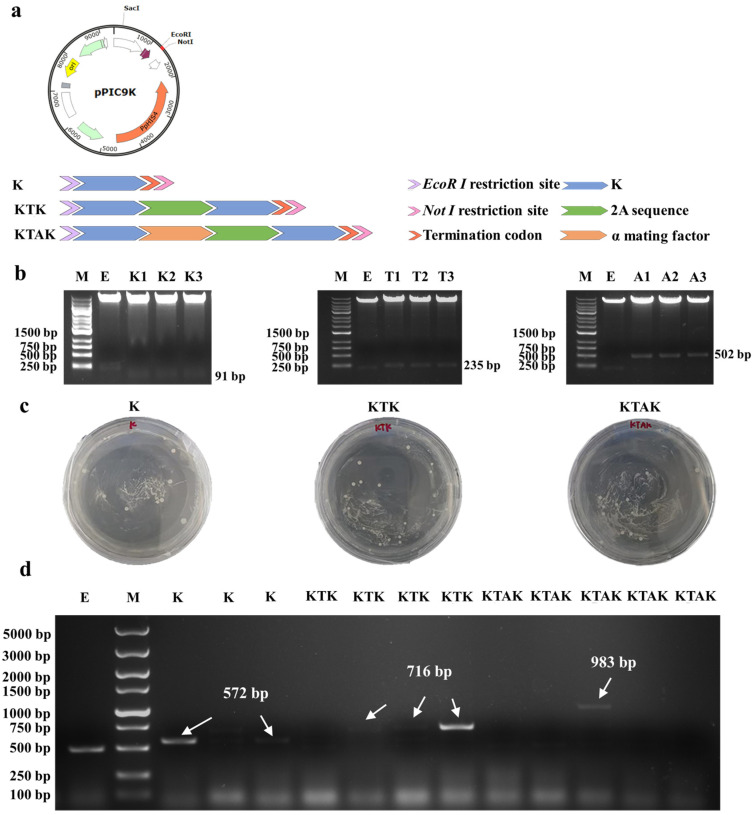
(**a**) Structure of pPIC9K plasmid and schematic diagram showing the alignment of K, KTK, and KTAK. (**b**) Agar gel electrophoresis for the recombinant plasmid. M: marker; E: pPIC9K; K: peptide K; T: KTK; A: KTAK. (**c**) Screening of positive transformants on MD plate. (**d**) PCR agar gel electrophoresis for monoclonal recombinant GS115 with universal primer *AOX*.

**Figure 2 antibiotics-13-00986-f002:**
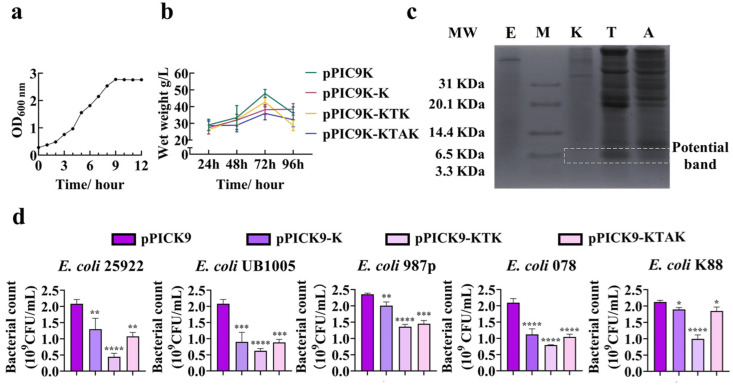
(**a**) Time growth of GS115. The culture parameters for the growth curve were a single colony of GS115 with empty vector in BMGY, growing at 30 °C in a shaking incubator 250 rpm. (**b**) Wet weight of recombinant GS115. (**c**) Tricine-SDS-PAGE for fermentation products from recombinant GS115. MW: molecular weight; E: supernatant from GS115 containing empty vector; K: supernatant from GS115 containing K; T: supernatant from GS115 containing KTAK; A: supernatant from GS115 containing KTAK. (**d**) The antibacterial activity of fermentation supernatant. * *p* < 0.05, ** *p* < 0.01; *** *p* < 0.001; **** *p* < 0.0001.

**Figure 3 antibiotics-13-00986-f003:**
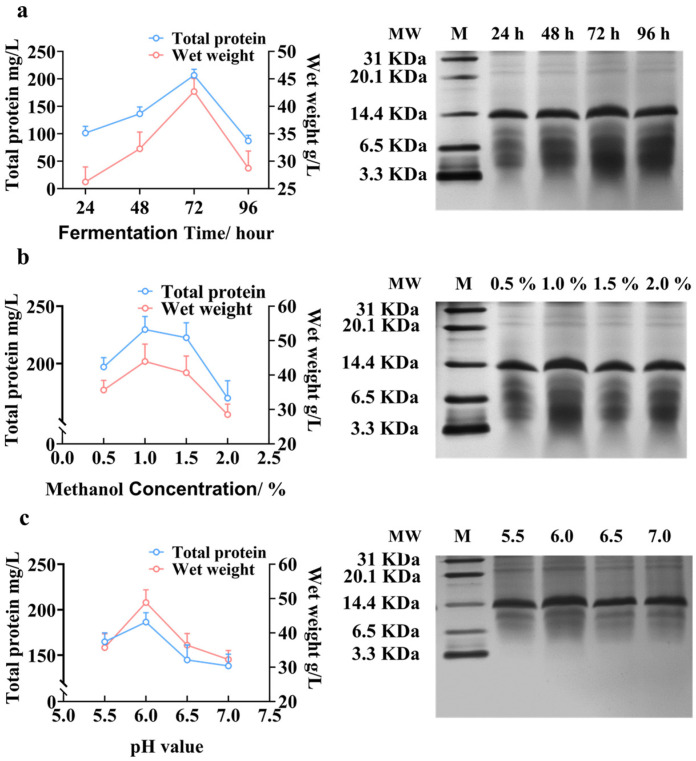
Expression of peptide K at different fermentation time (**a**) with different methanol concentration (**b**) and with different pH value (**c**). M: marker.

**Figure 4 antibiotics-13-00986-f004:**
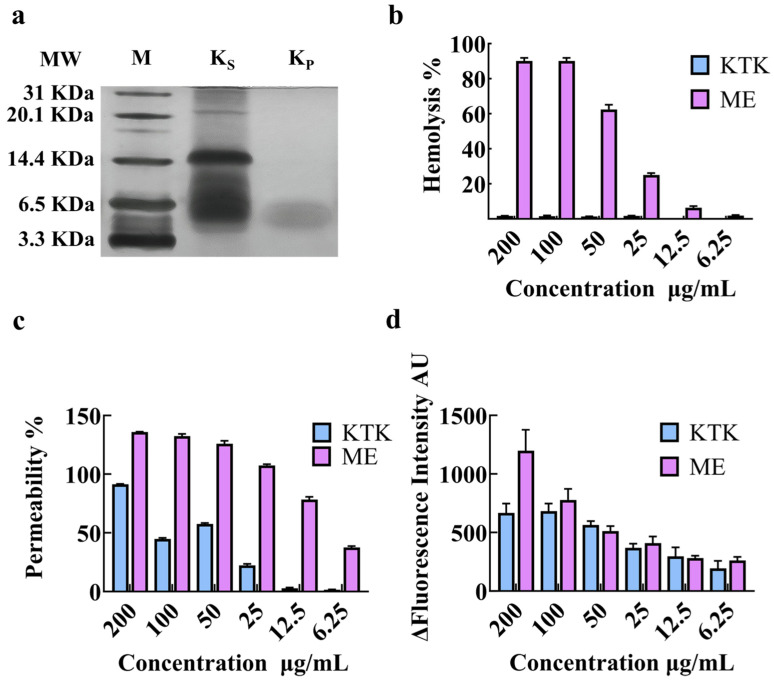
(**a**) Purification of recombinant peptide K. M: marker; K_S_: fermentation supernatant; K_P_: purified recombinant peptide K. (**b**) Hemolysis of purified peptide K from KTK. (**c**) Membrane permeability of recombinant peptide K using NPN method. (**d**) ROS production induced by recombinant peptide K using DCFH method. ME: melittin.

**Table 1 antibiotics-13-00986-t001:** The amino acid sequence of fragments for plasmid construction.

Items	Amino Acid Sequence
K	WKKIWKPGIKKWIKGGGQKRPRVRLSA
2A	EGRGSLLTCGDVEENPGP
α mating factor	MRFPSIFTAVLFAASSALAAPVNTTTEDETAQIPAEAVIGYSDLEGDFDVAVLPFSNSTNNGLLFINTTIASIAAKEEGVSLEKREAEA

**Table 2 antibiotics-13-00986-t002:** The production by recombinant GS115.

Item	Total, mg/L	Target, mg/L
pPIC9K-K	37.15	3.43
pPIC9K-KTK	92.99	18.31
pPIC9K-KTAK	55.38	2.92

‘Total’ was the concentration of total protein in supernatant determined using BCA method. ‘Target’ was the concentration of purified protein in supernatant calculated with (1) concentration of purified protein in elution buffer measured using BCA method, (2) volume of the elution buffer containing purified product, and (3) volume of supernatant used for obtaining purified product. Formula used is as below.

**Table 3 antibiotics-13-00986-t003:** The antimicrobial activity of purified recombinant peptide K.

Bacteria		MIC, μg/mL
Gram-negative bacteria	*E. coli* 25922	25
	*E. coli* UB1005	6.25
	*E. coli* K88	12.5
	*E. coli* K99	6.25
	*E. coli* 078	25
	*E. coli* 987p	12.5
	*P. aeruginosa* 27853	>400
	*S. typhimurium* 7731	>400
Gram-positive bacteria	*S. aureus* 29213	>400
	*S. aureus* 25923	>400
	*S.epidermidis* 12228	>400
Probiotics	*L. plantarum* 8014	>400
	*L. rhamnosus* 7469	>400

MIC: minimum inhibitory concentration.

**Table 4 antibiotics-13-00986-t004:** Antimicrobial activity of recombinant peptide K in the presence of different salts, pH, high temperature, and serum.

Salt ^1^				
	NaCl	KCl	NH_4_Cl	MgCl_2_
MIC, μg/mL	>400	50	50	200
	CaCl_2_	ZnCl_2_	FeCl_3_	
MIC, μg/mL	50	50	50	
pH				
	2	4	10	12
MIC, μg/mL	50	25	12.5	12.5
Temperature, °C				
	100			
MIC, μg/mL	12.5			
Serum, %				
	12.5	25	50	
MIC, μg/mL	100	100	100	

^1^ The concentration of NaCl, KCl, CaCl_2_, MgCl_2_, FeCl_3_, ZnCl_2_, and NH_4_Cl, were 150 mM, 4.5 mM, 2.5 mM, 1 mM, 4 μM, 8 μM, and 6 μM. MIC: minimum inhibitory concentration.

## Data Availability

All data supporting reported literature can be found in this paper.

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
