# Peer review of "Boosting Expression of a Specifically Targeted Antimicrobial Peptide K in Pichia pastoris by Employing a 2A Self-Cleaving Peptide-Based Expression System"

_antibiotics, 2024, doi:10.3390/antibiotics13100986_

Round 1
Reviewer 1 Report
Comments and Suggestions for Authors
The authors proposed a new strategy to recombinantly produce the antimicrobial peptide STAMP K using Pichia pastoris. They optimized the expression vector design and fermentation condition to achieve high yield. The topic of this manuscript fits the scope of antibiotics, but the authors will need to address the following concerns before I can recommend its publication.
1. Fig 2c: this gel image was presented poorly. It seems that the authors wish to show recombinant protein expression in cell lysates, but there was no clear labeling of any potential target band of their target protein. The fact that readers were never informed of the exact sizes of the three recombinant proteins designed in this manuscript (K, KTK, and KTAK) made the image more confusing. The authors should do mass spectrometry on their purified products to demonstrate they actually obtained the designed protein.
2. Fig 2a: It is unclear why the authors only provided the growth curve of the first 12 hours of fermentation, while the fermentation clearly lasted much longer. The authors should provide a complete growth curve (OD600), together with key fermentation parameters (temperature, pH, DO, agitation, feed, etc.) over the entire fermentation process.
3. Table 2 is very confusing. What experimental conditions were used in the fermentation process when the authors got these numbers? Are these proteins purified?
4. Fig 4a is greatly overexposed. The authors should reduce the brightness of the image to a reasonable value, or provide the original image. This is important because potential impurities on the gel may have been outshined by the bright background.
5. Table 3: The >400 ug/mL MIC presented for key strains such as S. aureus was not satisfying. Can authors provide a more precise number, or does STAMP K have no antimicrobial activity on these bacteria? The author should include previously reported MIC values in this table to compare the effectiveness of their recombinant protein.
6. Page 5, Line 149: The statement is confusing. The concentration of purified peptide K can be any value below its solubility, depending on the volume of the solvent. The 6.67 mg/mL number is meaningless. The author should report the total volume of cell culture in their fermentation and the total amount of purified protein they obtained from it.
7. The authors should create another table comparing previously reported protein yields (from chemical synthesis or bacterial expression, etc.) with theirs to show why using Pichia pastoris is a good strategy.
Comments on the Quality of English LanguageCan be improved through some professional editing, but overall fine and comprehensible.
Author Response
Dear reviewer,
On behalf of my co-authors, we thank you sincerely for giving us an opportunity to revise our manuscript. We are grateful for your positive comments and professional advices. The reviewer suggested moderate editing of the English language, and improvement in design of research, description of method, presentation of result, and support of conclusions. We agree with the reviewer, your comments help improve the academic rigor of our article. We have considered the comments very carefully. Based on your comments, we have revised the manuscript. We hope that our work can be improved again. The point-by-point response were in the attachment. Please see the attachment.
Best Wishes,
Na Dong,

Reviewer 2 Report
Comments and Suggestions for Authors
In this study, the authors constructed three different combinations of inserts based on a previously discovered antimicrobial peptide K onto pPIC9K vector and expressed these antimicrobial peptides from Pichia pastoris. The yield of the three peptides was analyzed and compared. The fermentation conditions were screened and optimized for KTK, which has the highest yield among the three peptide candidates. A series of assessments was conducted to study the selectivity, antibacterial activity, biocompatibility, and stability of fermentation product of KTK product. The results are of interest to the field of AMPs with potential application to current epidemic of drug-resistance bacterial strain issues. However, there are a few issues needed to be clarified before the manuscript is ready to be accepted for publication.
1. Why do the authors select E. coli 25922, UB1005, 987p, 078, and K88 as the test strains in the assessment of the antimicrobial activity of fermentation supernatants in 2.2? Are those E. coli strains the representative models of testing antimicrobial activity or just randomly picked for this experiment?
2. In the Figure 2c, even though the authors claimed that the 3.3 kDa band (peptide K) was failed to observe and provided a plausible explanation by comparing the smears of empty pPIC9K plasmid expression with other pPIC9K plasmid that contains target inserts, a more convincible characterization is required. This may be achieved by using mass spectrometry or WB with specific antibodies that can probe the peptide K, or any other equivalent means with direct evidence.
3. Similarly as the above, in the SDS-PAGE results in Figure 3, it is difficult to interpret the SDS-PAGE results if we do not know which band or smear is to be compared among different conditions. Perhaps the authors can put more descriptive information into this.
4. Additional information is needed to explain the purpose of melittin in assessing the bioactivity and actional mechanism of recombinant peptide K.
5. Since the STAMPs target K, KTK, and KTAK can all be purified via Ni-NTA column chromatography, I suggest the authors apply the purified K, KTK, and KTAK to different E. coli strains to study the antimicrobial activity. In this way, less impurities can interfere with the antimicrobial process.
Author Response
Dear reviewer,
On behalf of my co-authors, we thank you sincerely for giving us an opportunity to revise our manuscript. We appreciate reviewer very much for your thoughtful and constructive comments for our manuscript entitled “Boosting expression of a specifically targeted antimicrobial peptide K in Pichia pastoris by employing a 2A self-cleaving peptide-based expression system” (Manuscript ID: antibiotics-3231475). The reviewer suggested improvement in introduction and presentation of results. We have considered the comments very carefully and revised the paper accordingly. We hope that the revisions successfully address concerns of the reviewer and that this manuscript will be accepted. Looking forward to hearing from you soon. The point-by-point response were in the attachment. Please see the attachment.
Best Wishes,
Na Dong,

Reviewer 3 Report
Comments and Suggestions for Authors
The manuscript entitled "Boosting expression of a specifically targeted antimicrobial peptide K in Pichia pastoris by employing a 2A self-cleaving peptide-based expression system" is an interesting study that demonstrates the application of recombinant DNA technology towards the production of Antimicrobial Peptide. However, the quality of the manuscript can be further improved by addressing the following queries:
1. Authors should discuss the hurdles of expression system when connecting a Prokaryotic gene to a eukaryotic expression system.
2. The statement in lines 83-85 may be moved to after the conclusion statement in line 370.
3. Scientific names (e.g., in lines 263 and 278) should be italicized throughout the manuscript.
4. In Section 4.6, authors should clarify whether the same treatments of protocol were used for different bacteria studies during antibacterial activity. If different, please mention in this section.
5. In Figure 1b, the agar gel electrophoresis for the recombinant plasmid showing 91 bp is not matching. Authors may provide the correct figure.
6. Details of media optimization may be mentioned, particularly when RSM or other equivalent methods are used.
7. In the Materials and Methods section, details of protein estimation methods are missing. The same may be discussed for better clarity.
8. In the Materials and Methods section, details of protein purification may be mentioned for better clarity.
9. References are not written uniformly. Hence it may be written as per the author guidelines.
Author Response
Dear reviewer,
On behalf of my co-authors, we thank you sincerely for giving us an opportunity to revise our manuscript entitled “Boosting expression of a specifically targeted antimicrobial peptide K in Pichia pastoris by employing a 2A self-cleaving peptide-based expression system” (Manuscript ID: antibiotics-3231475).The reviewer suggested improvement in introduction and methodology. We agree with the reviewer, your comments help improve the academic rigor of our article. We have considered the comment very carefully and revised the paper accordingly. We hope that the revisions successfully address concerns of the reviewer and that this manuscript will be accepted. Looking forward to hearing from you soon.
Best Wishes,
Na Dong,

Round 2
Reviewer 1 Report
Comments and Suggestions for Authors
The authors have addressed my concerns, and I recommend the publication of the revised manuscript.
Reviewer 2 Report
Comments and Suggestions for Authors
The authors' response has genuinely explained the issues in detail and resolved all my concerns. I therefore am satisfied with the revised version of the manuscript and would like to recommend the article to be accepted.